# Understanding Isomorphism Bias in Graph Data Sets

## Abstract

In recent years there has been a rapid increase in classification methods on graph structured data. Both in graph kernels and graph neural networks, one of the implicit assumptions of successful state-of-the-art models was that incorporating graph isomorphism features into the architecture leads to better empirical performance. However, as we discover in this work, commonly used data sets for graph classification have repeating instances which cause the problem of isomorphism bias, i.e. artificially increasing the accuracy of the models by memorizing target information from the training set. The problem does not vanish even in consideration of node labels. This prevents fair competition of the algorithms and raises a question of the validity of the obtained results. First, we characterize this effect theoretically by reducing the graph classification to a weighted classification problem and estimating the corresponding generalization gap. Then we analyze 54 data sets, previously extensively used for graph-related tasks, on the existence of isomorphism bias, give a set of recommendations to machine learning practitioners to properly set up their models, and open source new data sets for the future experiments.

## 1 Introduction

Recently there has been an increasing interest in the development of machine learning models that operate on graph structured data. Such models have found applications in chemoinformatics (Ralaivola et al. (2005); Rupp & Schneider (2010); Ferré et al. (2017)) and bioinformatics (Borgwardt et al. (2005); Kundu et al. (2013)), neuroscience (Sharaev et al. (2018); Jie et al. (2016); Wang et al. (2016)), computer vision (Stumm et al. (2016)) and system security (Li et al. (2016)), natural language processing (Glavaš & Šnajder (2013)), and others (Kriege et al. (2019); Nikolentzos et al. (2019)). One of the popular tasks that encompasses these applications is graph classification problem for which many graph kernels and graph neural networks have been developed.

One of the implicit assumptions that many practitioners adhere to is that models that can distinguish isomorphic instances from non-isomorphic ones possess higher expressiveness in classification problem and hence much efforts have been devoted to incorporate efficient graph isomorphism methods into the classification models. As the problem of computing complete graph invariant is $\mathbb{GI}$-hard (Gärtner et al. (2003)), for which no known polynomial-time algorithm exists, other heuristics have been proposed as a proxy for deciding whether two graphs are isomorphic. Indeed, from the early days topological descriptors such Wiener index (Wiener (1947a;b)) attempted to find a single number that uniquely identifies a graph. Later, graph kernels that model pairwise similarities between graphs utilized theoretical developments in graph isomorphism literature. For example, graphlet kernel (Shervashidze et al. (2009)) is based on the Kelly conjecture (see also Kelly (1957)), anonymous walk kernel (Ivanov & Burnaev (2018)) derives insights from the reconstruction properties of anonymous experiments (see also Micali & Allen Zhu (2016)), and WL kernel (Shervashidze et al. (2011a)) is based on an efficient graph isomorphism algorithm. For sufficiently large $k$, $k$-dimensional WL algorithm includes all combinatorial properties of a graph (Cai et al. (1992a)), so one may hope its power is enough for the data set at hand. Since only for $k = \Omega(n)$ WL algorithm is guaranteed to distinguish all graphs (for which the running time becomes exponential; see also Fürer (2017)), in the general case WL algorithm can be used only as a strong baseline for graph isomorphism. In similar fashion, graph neural networks exploit graph isomorphism algorithms and

have been shown to be as powerful as k-dimensional WL algorithm (see for example Maron et al. (2019); Xu et al. (2018); Morris et al. (2019)).

Experimental evaluation reveals that models based on the theoretical constructions with high combinatorial power such as WL algorithm performs better than the models without them such as Vertex histogram kernel (Vishwanathan et al. (2010)) on a commonly used data sets. This could add additional bias to results of comparison of classification algorithms since the models could simply apply a graph isomorphism method (or an efficient approximation) to determine a target label at the inference time. However, purely judging on the accuracy of the algorithms in such cases would imply an unfair comparison between the methods as it does not measure correctly generalization ability of the models on the new test instances. As we discover, indeed many of the data sets used in graph classification have isomorphic instances so much that in some of them the fraction of the unique non-repeating graphs is as low as 20% of the total size. This challenges previous experimental results and requires understanding of how influential isomorphic instances on the final performance of the models. Our contributions are:

- We analyze the quality of 54 graph data sets which are used ubiquitously in graph classification comparison. Our findings suggest that in the most of the data sets there are isomorphic graphs and their proportion varies from as much as 100% to 0%. Surprisingly, we also found that there are isomorphic instances that have different target labels suggesting they are not suitable for learning a classifier at all.

- We investigate the causes of isomorphic graphs and show that node and edge labels are important to identify isomorphic graphs. Other causes include numerical attributes of nodes and edges as well as the sizes of the data set.

- We express an upper bound for the generalization gap through the Radamacher complexity of a classifier and the number of isomorphic graphs in a data set. This bound presents theoretical evidence on how weightning of each graph in the training influences classification accuracy.

- We evaluate a classification model's performance on isomorphic instances and show that even strong models do not achieve optimal accuracy even if the instances have been seen at the training time. Hence we show a model-agnostic way to artificially increase performance on several widely used data sets.

- We open-source new cleaned data sets that contain only non-isomorphic instances with no noisy target labels. We give a set of recommendations regarding applying new models that work with graph structured data.

## 2 RELATED WORK

**Measuring quality of data sets.** A similar issue of duplicates instances in commonly used data sets was recently discovered in computer vision domain. Recht et al. (2019); Barz & Denzler (2019); Birodkar et al. (2019) discover that image data sets CIFAR and ImageNet contain at least 10% of the duplicate images in the test test invalidating previous performance and questioning generalization abilities of previous successful architectures. In particular, evaluating the models in new test sets shows a drop of accuracy by as much as 15% (Recht et al., 2019), which is explained by models' incapability to generalize to unseen slightly "harder" instances than in the original test sets. In graph domain, a fresh look into understanding of expressiveness of graph kernels and the quality of data sets has been considered in Kriege et al. (2019), where an extensive comparison of existing graph kernels is done and a few insights about models' behavior are suggested. In contrast, we conduct a broader study of isomorphism metrics, revealing all isomorphism pairs in proposed 54 data sets, and propose new cleaned data. Additionally we also consider graph neural network performance and argue that current data sets present isomorphism bias which can artificially boost evaluation metrics in a model-agnostic way.

**Explaining performance of graph models.** Graph kernels (Kriege et al. (2019)) and graph neural networks (Wu et al. (2019)) are two competing paradigms for designing graph representations and solving graph classification and have significantly advanced empirical results due to more efficient algorithms, incorporating graph invariance into the models, and end-to-end training. Several papers have tried to justify performance of different families of methods by studying different statistical

properties. For example, in Ying et al. (2019) by maximizing mutual information between explanation variables and predicted label distribution, the model is trained to return a small subgraph and the graph-specific attributes that are the most influential on the decision made by a GNN, which allows inspection of single- and multi-level predictions in an agnostic manner for GNNs. In another work (Scarselli et al. (2018)), the VC dimension of GNNs models has been shown to grow as $\mathcal{O}(p^4N^2)$, where $p$ is the number of network parameters and $N$ is the number of nodes in a graph, which is comparable to RNN models. Furthermore, stability and generalization properties of convolutional GNNs have been shown to depend on the largest eigenvalue of the graph filter and therefore are attained for properly normalized graph convolutions such as symmetric normalized graph Laplacian (Verma & Zhang (2019)). Finally, expressivity of graph kernels has been studied from statistical learning theory (Oneto et al. (2017b)) and property testing (Kriege et al. (2018b)), showing that graph kernels can capture certain graph properties such as planarity, girth, and chromatic number (Johansson et al. (2014)). Our approach is complementary to all of the above as we analyze if the data sets used in experiments have any effect on the final performance.

## 3  PRELIMINARIES

In this work we analyze 54 graph data sets from Kersting et al. (2016) that are commonly used in graph classification task. Examples of popular graph data sets are presented in Table 1 and statistics of all 54 data sets can be found in Table 5, see Section A in the appendix. All data sets represent a collection of graphs and accompanying categorical label for each graph in the data sets. Some data sets also include node and/or edge labels that graph classification methods can use to improve the scoring. Most of the data sets come either from biological domain or from social network domain. Biological data sets such as MUTAG, ENZYMES, PROTEINS are graphs that represent small or large molecules, where edges of the graphs are chemical bonds or spatial proximity between different atoms. Graph labels in these cases encode different properties like toxicity. In social data sets such as IMDB-BINARY, REDDIT-MULTI-5K, COLLAB the nodes represent people and edges are relationships in movies, discussion threads, or citation network respectively. Labels in these cases denote the type of interaction like the genre of the movie/thread or a research subfield. For completeness we also include synthetic data sets SYNTHETIC (Morris et al. (2016)) that have continuous attributes and computer vision data sets MSRC (Neumann et al. (2016)), where images are encoded as graphs. The origin of all data sets can be found in the Table 5.

Table 1: Example of graph data sets. $N$ is the number of graphs, $C$ is the number of different classes. Avg. Nodes and Avg. Edges is the average number of nodes and edges. N.L. and E.L indicate if the graphs in a data set contain node or edge labels.

| data set | Type | $N$ | $C$ | Avg. Nodes | Avg. Edges | N.L. | E.L. |
|---|---|---|---|---|---|---|---|
| MUTAG | Molecular | 188 | 2 | 17.93 | 19.79 | + | + |
| ENZYMES | Molecular | 600 | 6 | 32.63 | 62.14 | + | - |
| PROTEINS | Molecular | 1113 | 2 | 39.06 | 72.82 | + | - |
| IMDB-BINARY | Social | 1000 | 2 | 19.77 | 96.53 | - | - |
| REDDIT-MULTI-5K | Social | 4999 | 5 | 508.52 | 594.87 | - | - |
| COLLAB | Social | 5000 | 3 | 74.49 | 2457.78 | - | - |
| SYNTHETIC | Synthetic | 300 | 2 | 100 | 196 | - | - |
| Synthie | Synthetic | 400 | 4 | 95 | 172.93 | - | - |
| MSRC_21C | Vision | 209 | 20 | 40.28 | 96.6 | + | - |
| MSRC_9 | Vision | 221 | 8 | 40.58 | 97.94 | + | - |

**Graph isomorphism.** Isomorphism between two graphs $G_1 = (V_1, E_1)$ and $G_2 = (V_2, E_2)$ is a bijective function $\phi : V_1 \mapsto V_2$ such that any edge $(u, v) \in E_1$ if and only if $(\phi(u), \phi(v)) \in E_2$. Graph isomorphism problem asks if such function exists for given two graphs $G_1$ and $G_2$. We denote isomorphic graphs as $G_1 \cong G_2$. The problem has efficient algorithms in $\mathbb{P}$ for certain classes of graphs such as planar or bounded-degree graphs (Hopcroft & Wong (1974); Luks (1980)), but in the general case admits only quasi-polynomial algorithm (Babai (2015)). In practice many GI solvers are based on individualization-refinement paradigm (Mckay & Piperno (2014)), which for each graph iteratively updates a permutation of the nodes such that the resulted permutations of two

graphs are identical if an only if they are isomorphic. Importantly, while finding such canonical permutation of a graph is at least as hard as solving GI problem, state-of-the-art solvers tackle majority of pairs of graphs efficiently, only taking exponential time on the specific hard instances of graphs that possess highly symmetrical structures (Cai et al. (1992b)).

## 4    IDENTIFYING ISOMORPHISM IN DATA SETS

To distinguish between different isomorphic graphs inside a data set we use the notion of graph orbits:

**Definition 4.1** (Graph orbit). Let $\mathcal{D} = \{G_i, y_i\}_{i=1}^N$ be a data set of graphs and target labels. For a graph $G_i$ let a set $o_i = \{G_k\}$ be a set of all isomorphic graphs in $\mathcal{D}$ to $G_i$, including $G_i$. We call $o_i$ the *orbit* of graph $G_i$ in $\mathcal{D}$. The cardinality of the orbit is called *orbit size*. An orbit with size one is called *trivial*.

In a data set with no isomorphic graphs, the number of orbits equals to the number of graphs in a data set, $N$. Hence, the more orbits in a data set, the "cleaner" it is. Note however that the distribution of orbit sizes in two different data sets can vary even if they have the same number of orbits. Therefore, we look at additional metrics that describe the data set.

- $I$, aggregated number of graphs that belong to an orbit of size greater than one, i.e. those graphs that isomorphic counterparts in a data set;
- $I, \%$, proportion of isomorphic graphs to the total data set size, i.e. $\frac{I}{N}$;
- $IP, \%$, proportion of isomorphic pairs to the total number of graph pairs in a data set $(\frac{N(N-1)}{2})$.

If we consider target labels of graphs in a data set $\mathcal{D} = \{G_i, y_i\}_{i=1}^N$ we can also measure agreement between the labels of two isomorphic data set. If $G_1 \cong G_2$ and $y_1 \neq y_2$, then we call graphs *mismatched*. Note that if there is more than one target label in an orbit $o$, then all graphs in this orbit are mismatched. To obtain isomorphic graphs, we run *nauty* algorithm (Mckay & Piperno, 2014) on all possible pairs of graphs in a data set. We substantially reduce the number of calls between the graphs by verifying that a pair has the same number of nodes and edges before the call.

The metrics are presented in Table 2 for top-10 data sets and in Table 6 (see the appendix) for all data sets. The graphs in Table 2 are sorted by the proportion of isomorphic graphs $I\%$. The results for the first Top-10 data sets are somewhat surprising: almost all graphs in the selected data sets have other isomorphic graphs. If we look at all data sets in Table 6, we see that the proportion of isomorphic graphs in the data sets varies from 100% to 0%. However, *more than 80% of the analyzed data sets have at least 10% of the graphs in a non-trivial orbit.*

Table 2: Isomorphic metrics for Top-10 data sets based on the proportion of isomorphic graphs $I\%$. $IP\%$ is the proportion of isomorphic pairs of graphs, Mismatched $\%$ is the proportion of mismatched labels.

| data set | Size, $N$ | Num. orbits | Iso. graphs, $I$ | $I\%$ | $IP\%$ | Mismatched $\%$ |
|---|---|---|---|---|---|---|
| SYNTHETIC | 300 | 2 | 300 | 100 | 100 | 100 |
| Cuneiform | 267 | 8 | 267 | 100 | 20.46 | 100 |
| Letter-low | 2250 | 32 | 2245 | 99.78 | 8.72 | 96.22 |
| DHFR_MD | 393 | 25 | 392 | 99.75 | 6.87 | 94.91 |
| COIL-RAG | 3900 | 20 | 3890 | 99.74 | 25.22 | 99.31 |
| COX2_MD | 303 | 13 | 301 | 99.34 | 11.83 | 98.68 |
| ER_MD | 446 | 31 | 442 | 99.1 | 5.57 | 82.74 |
| Fingerprint | 2800 | 69 | 2774 | 99.07 | 16.86 | 89.29 |
| BZR_MD | 306 | 22 | 303 | 99.02 | 7.16 | 95.75 |
| Letter-med | 2250 | 39 | 2226 | 98.93 | 8.05 | 92.93 |

Another surprising observation is that the proportion of mismatched graphs is significant, ranging from 100% to 0%. This clearly indicates that such graphs are not suitable for graph classification

and require additional information to distinguish the models. We analyze the reasons for this in the next section.

Also, the distribution of orbit sizes can vary significantly across the data sets. In Figure 1 we plot a distribution of orbit sizes for several examples of data sets (and distributions for other data sets can be found in Appendix C). For example, for IMDB-BINARY data set the number of orbits of small sizes, e.g. two or three, goes to 100, which indicate prevalence of pairs of isomorphic graphs that are non-isomorphic to the rest. However, for Letter-med data set there are many orbits of sizes more than 100, while small orbits are not that common. In this case, the graphs in this data set are equivalent to a lot of other graphs, which may have a substantial effect on the corresponding metrics. While the orbit distribution changes from one data set to another, it is clear that in many situations there are isomorphic graphs that can affect training procedure by effectively increasing the weights for the corresponding graphs, change performance on the test by validating on the already seen instances, and by confusing the model by utilizing different target labels for topologically-equivalent graphs. We analyze the reasons for it further.

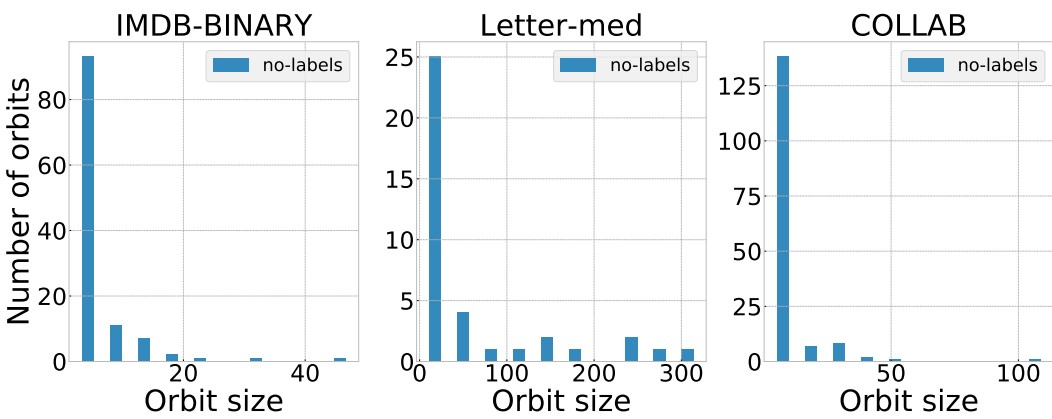

Figure 1: Examples of distributions of orbit sizes without considering labels.

## 5 EXPLAINING ISOMORPHISM

**Meta-information about graphs.** In addition to the topology of a graph, many data sets also include meta information about nodes and/or edges. Out of 54 analyzed data sets there are 40 that additionally include node features and 25 that include edge features. For example, in Synthetic data set all graphs are topologically identical but the nodes are endowed with normally distributed scalar attributes and in DHFR_MD edges are attributed with distances and labeled according to a chemical bond type. Alternatively, some graphs can have parallel edges which is equivalent to have a corresponding weight on the edges. Thus some data sets include node/edge categorical features (labels) and numerical features (attributes), which leads to better distinction between the graphs and therefore their corresponding labels.

To see this, we rerun our previous analysis but now include the node labels, if any, when computing isomorphism between graphs. Consider a tuple $(G, l)$, where $G$ is a graph and $l : V(G) \mapsto \{1, 2, \ldots, k\}$ is a $k$-labeling of $G$. In this case of node *label-preserving graph isomorphism* from graph $(G_1, l_1)$ to graph $(G_2, l_2)$ we seek an isomorphism function $\phi : V(G_1) \mapsto V(G_2)$ such that $l_1(v) = l_2(\phi(v))$.

Tables 3 and 7 (see the appendix) show the number of isomorphic graphs after considering node labels. While for the first six data sets the proportion of isomorphic graphs has not changed much, it is clearly the case for the remaining data sets. In particular, *almost 90% of the analyzed data sets include less than 20% of isomorphic graphs*. Also, the number of mismatched graphs significantly decreases after considering node labels. For example, for MUTAG data set the proportion of isomorphic graphs went down from 42.02% to 19.15% and the proportion of mismatched graphs from 6.91% to 0%.

Table 3: Isomorphic metrics with node labels for Top-10 data sets based on the proportion of isomorphic graphs $I\%$. $IP\%$ is the proportion of isomorphic pairs of graphs, Mismatched $\%$ is the proportion of mismatched labels.

| data set | Size, $N$ | Num. orbits | Iso. graphs, $I$ | $I\%$ | $IP\%$ | Mismatched $\%$ |
|---|---|---|---|---|---|---|
| SYNTHETIC | 300 | 2 | 300 | 100 | 100 | 100 |
| Cuneiform | 267 | 8 | 267 | 100 | 20.46 | 100 |
| DHFR_MD | 393 | 25 | 392 | 99.75 | 6.87 | 94.91 |
| COX2_MD | 303 | 13 | 301 | 99.34 | 11.83 | 98.68 |
| ER_MD | 446 | 31 | 442 | 99.1 | 5.57 | 82.74 |
| BZR_MD | 306 | 22 | 303 | 99.02 | 7.16 | 95.75 |
| MUTAG | 188 | 17 | 36 | 19.15 | 0.14 | 0 |
| PTC_FM | 349 | 22 | 54 | 15.47 | 0.08 | 10.89 |
| PTC_MM | 336 | 22 | 50 | 14.88 | 0.07 | 7.74 |
| DHFR | 756 | 39 | 98 | 12.96 | 0.04 | 3.97 |

Likewise, the orbit size distribution also changes significantly after considering node labels. Figure 2 shows a changed distribution of orbits with and without considering node labels. For majority of data sets large orbits vanish and the number of small orbits is substantially decreased in label-preserving graph isomorphism setting. This indicates one of the reasons for presence of many isomorphic graphs in the data sets, which implies that including node/edge labels/attributes can be important for graph classification models.

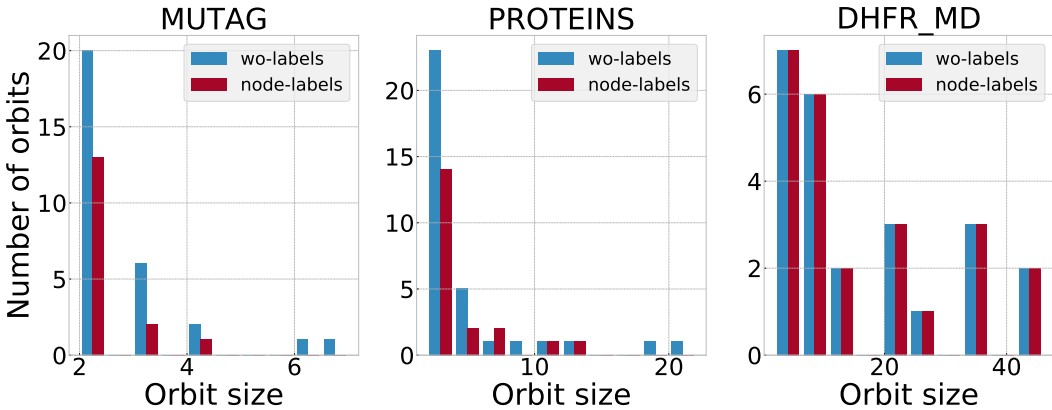

Figure 2: Examples of distributions of orbit sizes with node labels.

**Sizes of the data sets.** Another reason for having isomorphism in a data set is the sizes of graphs, which could be too small on average to lead to a diversity in a data set. In general, the number of non-isomorphic graphs with $n$ vertices and $m$ edges can be computed using Polya enumeration theory and grows very fast. For example, for a graph with 15 nodes and 15 edges, there are 2,632,420 non-isomorphic graphs. Nevertheless, specifics of the origin of the data set may affect possible configurations that graphs have (e.g. structure of chemical compounds in COX2_MD or ego-networks for actors in IMDB-BINARY) and thus smaller graphs may tend to be close to isomorphic structures. On the other hand, all five data sets with the average number of nodes greater than 100 have very low or zero proportion of isomorphic graphs. Hence, the average size of the graphs directly impacts the possible structure of the data set and thus data sets with larger graphs tend to be more diverse. We next analyze the consequences of the isomorphic graphs on classification methods.

## 6 ISOMORPHIC GRAPHS AND WEIGHTED CLASSIFICATION

We denote by

- $\mathcal{F} \subseteq Y^X$ a class of binary classifiers with an input space $X$ and an output space $Y = \{-1, +1\}$,

- $\mathbb{P}$ a distribution on $X \times Y$,

- $\pi$ a prior probability of a positive class, i.e. $\mathbb{P} = \pi \mathbb{P}_{x|y=+1} + (1-\pi)\mathbb{P}_{x|y=-1}$,

- $\mathcal{D} = \{(x_i, y_i)\}_{i=1}^N$ a training sample, $x_i \in X$, $y_i \in Y$. In our case $x_i \in \mathcal{D}$ are graphs $G_i$, see notations in section 4.

We consider a zero-one loss function $l(\hat{y}, y) = \mathbb{I}_{\hat{y} \neq y}$. Let us assume that classifiers from $\mathcal{F}$ can detect which graphs in $\mathcal{D}$ are isomorphic. E.g. classifiers based on Weisfeiler-Lehman graph kernels (see Shervashidze et al. (2011a)) are capable to do it for majority of graphs. In such case the empirical risk $\mathbb{E}_{\mathcal{D}} l(f(x), y) = \frac{1}{N} \sum_{j=1}^N l(f(x_j), y_j)$ reduces to $\frac{1}{N} \sum_{j \in J} u_j l(f(x_j), y_j)$, where $J$ is an index set of non-isomorphic graphs, $u_j \geq 1$ is equal to the number of graphs in the initial sample $\mathcal{D}$, isomorphic to the graph $x_j$ (we count $x_j$ as well).

Thus under such assumptions the graph classification problem with the training data set, containing isomorphic graphs, can be interpreted as a classification problem with a weighted loss. Let us introduce general notations for this problem. We define some (fixed) measurable weighting function $u : (X \times Y) \to (0, +\infty)$. Then the theoretical risk is equal to $\mathbb{E}_{\mathbb{P}} l(f(x), y)$ and the weighted empirical risk is equal to $\mathbb{E}_{\mathcal{D}} u(x, y) l(f(x), y) = \frac{1}{N} \sum_{i=1}^N u(x_i, y_i) l(f(x_i), y_i)$. We would like to derive an upper bound for the excess risk $\sup_{f \in \mathcal{F}} (\mathbb{E}_{\mathbb{P}} l(f(x), y) - \mathbb{E}_{\mathcal{D}} u(x, y) l(f(x), y))$, i.e. we would like to quantify an upper bound for a generalization gap. We optimize the weighted empirical risk when training a classifier and measure its accuracy using non-weighted theoretical risk.

There are some results about classification performance with a weighted loss. E.g. in (Dupret & Koda, 2001) a bayesian framework for imbalanced classification with a weighted risk is proposed. Scott (2012) investigated the calibration of asymmetric surrogate losses. Natarajan et al. (2018) considered the case of cost-sensitive learning with noisy labels. However, to the best of our knowledge, there is no studied upper bound for the excess risk with explicit dependence on the class imbalance and the weighting scheme that quantifies the influence on the overall classification performance. We show this result next.

To derive explicit expressions we use some additional modeling asumption, namely, we consider $u(x, y) = (1 + g_+(w))\mathbb{I}_{\{y=+1\}} + (1 + g_-(w))\mathbb{I}_{\{y=-1\}}$ for some non-negative weighting functions $g_+(w)$ and $g_-(w)$ of the weight value $w \geq 0$. E.g. we can use $g_+(w) = w$ and $g_-(w) = 1/w$.

**Theorem 6.1.** With probability $1 - \delta$, $\delta > 0$ for $\mathcal{D} \sim \mathbb{P}^N$ the excess risk is upper bounded by

$$\sup_{f \in \mathcal{F}} \left( \mathbb{E}_{\mathbb{P}} l(f(x), y) - \mathbb{E}_{\mathcal{D}} u(x, y) l(f(x), y) \right) \leq 3 \left( g_+(w)\pi + g_-(w)(1-\pi) \right) +$$

$$+ \mathcal{R}_N(\mathcal{F}) + (2 + g_+(w) + g_-(w)) \sqrt{(\log \delta^{-1})/(2N)}, \quad (1)$$

where $\mathcal{R}_N(\mathcal{F})$ is a Rademacher complexity of the function class $\mathcal{F}$.

Let us note that the Rademacher complexity of the function class $\mathcal{F}$, defined by a graph kernel, was studied e.g. in Oneto et al. (2018); Oneto et al. (2017a).

From equation 1 it follows that by tuning the weight parameter $w$ we can make the upper bound tighter, namely collecting the terms with $w$ in the RHS of equation 1 we solve

$$g_+(w)\left(3\pi + \sqrt{(\log \delta^{-1})/(2N)}\right) + g_-(w)\left(3(1-\pi) + \sqrt{(\log \delta^{-1})/(2N)}\right) \to \min_w.$$

In case we set $g_+(w) = w$ and $g_-(w) = 1/w$, the optimal weight $w_{opt} = \sqrt{\frac{3(1-\pi)+\alpha_N}{3\pi+\alpha_N}} \approx \sqrt{\frac{1-\pi}{\pi}}$, where $\alpha_N = \sqrt{\frac{\log \delta^{-1}}{2N}} \approx 0$ for $N \gg 1$. For such optimal $w_{opt}$ the RHS of equation 1 has the form

$$6\sqrt{\pi(1-\pi)} + \mathcal{R}_N(\mathcal{F}) + \alpha_N \left(2 + [\pi(1-\pi)]^{-1/2}\right).$$

Thus we get theoretical evidence on how the weighting influences the classification accuracy: e.g. in the imbalanced case (when $\pi \approx 0$ or $\pi \approx 1$) selecting the weight optimally we reduce the generalization gap almost to zero for $N \gg 1$; at the same time, not optimal weight can lead to overfitting.

As we already discussed, under some mild modeling assumptions the graph classification problem with isomorphic graphs in the training data set can be interpreted as the classification problem with a weighted loss. Therefore the obtained estimate provides additional evidence on a negative effect of the isomorphic graphs when solving the graph classification problems: the presence of isomorphic graphs in the training data set could have the same negative effect as not optimal weight value for the classification with a weighted loss function.

## 7 INFLUENCE OF ISOMORPHISM BIAS: EMPIRICAL RESULTS

To understand the impact of isomorphic graphs in the data set on the final metric we consider separately the results on two subparts of the data set. In particular, let $Y_{train}$ and $Y_{test}$ be train and test splits of a data set. Let $H_i \in Y_{test}$ be a graph such that there exists an isomorphic graph $G_i \in Y_{train}$ in the train data set. Let $\{Y_{iso}\}$ be a set of all such graphs $H_i$ for which there exists an isomorphic graph $G_i$ in $Y_{train}$. Note that the graphs in $\{Y_{iso}\}$ are not necessarily isomorphic. We denote by $Y_{new}$ the test graphs that do not have isomorphic copies in the train data set, i.e. $Y_{new} = Y_{test} \setminus Y_{iso}$. If we want to test generalization of classification models, we need to test it on new instances of the data sets and therefore at least consider $Y_{new}$ instead of $Y_{test}$. One question regarding the performance of the models on this new test set $Y_{new}$ is whether the performance on it will be lower than on the original test set $Y_{test}$. As we show below answer to this question solely depends on the accuracy of the model on isomorphic instances $Y_{iso}$.

Consider a graph classification model that is evaluated on normalized accuracy over a data set $Y$:

$$\mathrm{acc}(Y) = \frac{\sum\limits_{G \in Y} \mathrm{acc}(G_i)}{|Y|}, \tag{2}$$

where $\mathrm{acc}(G)$ equals to one if the model predicts the label of $G_i$ correctly, and zero otherwise. If $|Y| = 0$, then we consider $\mathrm{acc}(Y) = 0$. We can see that the accuracy on the test data set can be written as the sum of two terms:

$$\mathrm{acc}(Y_{test}) = \frac{\sum\limits_{G \in Y_{test}} \mathrm{acc}(G)}{|Y_{test}|} = \frac{\sum\limits_{G \in Y_{new}} \mathrm{acc}(G) + \sum\limits_{G \in Y_{iso}} \mathrm{acc}(G)}{|Y_{test}|} =$$
$$= \frac{|Y_{new}|}{|Y_{test}|} \mathrm{acc}(Y_{new}) + \frac{|Y_{iso}|}{|Y_{test}|} \mathrm{acc}(Y_{iso}). \tag{3}$$

Equation 3 decomposes accuracy on the original data set as the weighted sum of two accuracies on the set of the new test instances $Y_{new}$ and a set of the instances $Y_{iso}$ already appeared in the train set and therefore available to the model. We call the term $\mathrm{acc}(Y_{iso})$ as *isomorphism bias*, which corresponds to the accuracy of the model on the isomorphic test instances. As we will see next, the accuracy of the model on the new set $Y_{new}$ will be less if only if the model performs better on the isomorphic set $Y_{iso}$.[1]

**Claim 7.1.** Let $Y_{test} = Y_{new} \cup Y_{iso}$, $Y_{new} \cap Y_{iso} = \varnothing$ and $Y_{iso} \neq \varnothing$, where $Y_{iso} \subset Y_{train}$. Then for any classification model accuracy on the new test instances $Y_{new}$ is smaller than on the test set $Y_{test}$ if and only if it is smaller than accuracy on the isomorphic test instances $Y_{iso}$, i.e.:

$$\mathrm{acc}(Y_{test}) > \mathrm{acc}(Y_{new}) \iff \mathrm{acc}(Y_{iso}) > \mathrm{acc}(Y_{new}) \tag{4}$$

The equation 4 gives a definite answer with the possible performance of the model on a new test set. If the model performs well on isomorphic instances $Y_{iso}$, then it will falsely increase performance on $Y_{test}$ in comparison to $Y_{new}$. Conversely, if the model performs poorly on the instances that appeared in the training set, then removing them from the test set and evaluating the model purely on $Y_{new}$ will demonstrate higher accuracy. There are two reasons for the model to misclassify isomorphic instances $Y_{iso}$: (i) the instances contain target labels that are different than those that it has seen, as we show in Table 2 the percentage of mismatched labels can be high in some data

---

[1]We provide the proof of Claim 7.1 and Claim 7.2 in Appendix E and F.

Table 4: Mean classification accuracy for test sets $Y_{test}$ and $Y_{iso}$ (in brackets) in 10-fold cross-validation. Top-1 result in bold.

|       | MUTAG         | IMDB-B            | IMDB-M            | COX2          | AIDS              | PROTEINS          |
|-------|---------------|-------------------|-------------------|---------------|-------------------|-------------------|
| NN    | 0.829 (0.840) | 0.737 (0.733)     | 0.501 (0.488)     | 0.82 (0.872)  | 0.996 (0.998)     | 0.737 (0.834)     |
| NN-PH | 0.867 (1.000) | **0.756** (1.000) | **0.522** (1.000) | **0.838** (1.000) | **0.996** (1.000) | 0.742 (1.000) |
| NN-P  | 0.856 (0.847) | 0.737 (0.731)     | 0.499 (0.486)     | 0.795 (0.83)  | 0.996 (0.999)     | 0.729 (0.709)     |
| WL    | 0.862 (0.867) | 0.734 (0.990)     | 0.502 (0.953)     | 0.800 (0.974) | 0.993 (0.999)     | 0.747 (0.950)     |
| WL-PH | **0.907** (1.000) | 0.736 (1.000) | 0.504 (1.000)     | 0.810 (1.000) | 0.994 (1.000)     | **0.749** (1.000) |
| WL-P  | 0.870 (0.838) | 0.724 (0.715)     | 0.495 (0.487)     | 0.794 (0.844) | 0.994 (0.999)     | 0.740 (0.742)     |
| V     | 0.836 (0.902) | 0.707 (0.820)     | 0.503 (0.732)     | 0.781 (0.966) | 0.994 (0.997)     | 0.726 (0.946)     |
| V-PH  | 0.859 (1.000) | 0.750 (1.000)     | 0.517 (1.000)     | 0.794 (1.000) | **0.996** (1.000) | 0.729 (1.000)     |
| V-P   | 0.827 (0.844) | 0.724 (0.728)     | 0.496 (0.481)     | 0.768 (0.852) | 0.996 (0.999)     | 0.719 (0.741)     |

sets; or (ii) the model is not expressive enough to map the structure of the graphs to the target label correctly.

Crucially, while $Y_{new}$ tests generalization capabilities of the models, on $Y_{iso}$ the models can explicitly or implicitly memorize the right labels from the training. We describe a model-agnostic way to guarantee increase of classification performance if $|Y_{iso}| \neq 0$.

Let $G \cong \widehat{G}$ such that $G \in Y_{iso}$ and $\widehat{G} \in Y_{train}$. Note that there can be multiple isomorphic graphs $\{\widehat{G}_i\} \subset Y_{train}$. If for any $G \in Y_{iso}$ all target labels of the orbit of $G$ are the same we call the set $Y_{iso}$ as *homogeneous*. Consider a classification model $\mathcal{M}$ that maps each graph $G$ to its label $l(G)$. We define a *peering* model $\widehat{\mathcal{M}}$ such that for each $G \in Y_{iso}$ it outputs the target label $l(\widehat{G})$. Then the accuracy of the model $\widehat{\mathcal{M}}$ is at least as the accuracy of the original model $\mathcal{M}$.

**Claim 7.2.** Let $Y_{test} = Y_{new} \cup Y_{iso}$, $Y_{new} \cap Y_{iso} = \varnothing$. If $Y_{iso}$ is homogeneous, then the accuracy on $Y_{test}$ of a classification model $\mathcal{M}$ is at most as the accuracy of its peering model $\hat{\mathcal{M}}$, i.e.:

$$\text{acc}_{\mathcal{M}}(Y_{test}) \leq \text{acc}_{\widehat{\mathcal{M}}}(Y_{test}).$$

Claim 7.2 establishes a way to increase performance only for homogeneous $Y_{iso}$. If there are noisy labels in the training set and hence the set is not homogeneous, the model cannot guarantee the right target label for these instances. Nonetheless, one can select a heuristic such as majority vote among the training isomorphic instances to select a proper label at the testing time.

In experiments, we compare neural network model (**NN**) (Xu et al., 2018) with graph kernels, Weisfeiler-Lehman (**WL**) (Shervashidze et al., 2011b) and vertex histogram (**V**) (Sugiyama & Borgwardt, 2015). For each model we consider two modifications: one for peering model on homogeneous $Y_{iso}$ (e.g. **NN-PH**) and one for peering model on all $Y_{iso}$ (e.g. **NN-P**). We show accuracy on $Y_{test}$ and on $Y_{iso}$ (in brackets) in Table 4. Experimentation details can be found in Appendix G.

From Table 4 we can conclude that peering model on homogeneous data is always the top performer. This is aligned with the result of Claim 7.2, which guarantees that $acc(Y_{iso}) = 1$, but it is an interesting observation if we compare it to the peering model on all isomorphic instances $Y_{iso}$ (-P models). Moreover, the latter model often loses even to the original model, where no information from the train set is explicitly taken into the test set. This can be explained by the noisy target labels in the orbits of isomorphic graphs, as can be seen both from the statistics for these datasets (Table 6) and accuracy measured just on isomorphic instances $Y_{iso}$. These results show that due to the presence of isomorphism bias performance of any classification model can be overestimated by as much as 5% of accuracy on these datasets and hence future comparison of classification models should be estimated on $Y_{new}$ instead. These observations conforms with our theoretical findings and conclusions in Section 6.

## 7.1 GENERAL RECOMMENDATIONS

In order to avoid measuring performance over the wrong test sets, we provide a set of recommendations that will guarantee measuring the right metrics for the models.

- We open-source new, "clean" data sets that do not include isomorphic instances that are in Table 8. To tackle this problem in the future, we propose to use clean versions of the data set for which isomorphism bias vanishes. For each data set we consider the found graph orbits and keep only one graph from each orbit if and only if the graphs in the orbit have the same label. If the orbit contains more than one label, a classification model can do little to predict a correct label at the inference time and hence we remove such orbit completely. In this case, for a new data set $Y_{iso} = \varnothing$ and hence it prevents the models to implicitly memorize the labels from the training set. We consider the data set orbits that do not account for neither node nor edge labels because the remaining graphs are not isomorphic based purely on graph topology.

- Incorporating node and edge features into the models may be necessary to distinguish the graphs. As we have seen, just using node labels can reduce the number of isomorphic graphs significantly and many data sets provide additional information to distinguish the models at full scope.

- Verification of the models on bigger graphs in general is more challenging due to the sheer number of non-isomorphic graphs. For example, data sets related to REDDIT or DD include a number of big graphs for classification.

## 8 CONCLUSION

In this work we study isomorphism bias of the classification models in graph structured data that originates from substantial amount of isomorphic graphs in the data sets. We analyzed 54 graph data sets and provide the reasons for it as well as a set of rules to avoid unfair comparison of the models. We theoretically characterized the influence of isomorphism bias on the graph classification performance by providing an upper bound on the generalization gap. We showed that in the current data sets any model can memorize the correct answers from the training set and we open-source new clean data sets where such problems do not appear.

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

# A STATISTICS FOR ORIGINAL DATA SETS

Table 5: All original graph data sets. $N$ is the number of graphs, $C$ is the number of different classes. Avg. Nodes and Avg. Edges is the average number of nodes and edges. N.L. and E.L indicate if the graphs in a data set contain node or edge labels.

| data set | Type | $N$ | $C$ | Avg. Nodes | Avg. Edges | N.L. | E.L. | Source |
|---|---|---|---|---|---|---|---|---|
| FIRSTMM_DB | Molecular | 41 | 11 | 1377.27 | 3074.1 | + | - | Neumann et al. (2016) |
| OHSU | Molecular | 79 | 2 | 82.01 | 199.66 | + | - | Pan (2018) |
| KKI | Molecular | 83 | 2 | 26.96 | 48.42 | + | - | Pan (2018) |
| Peking_1 | Molecular | 85 | 2 | 39.31 | 77.35 | + | - | Pan (2018) |
| MUTAG | Molecular | 188 | 2 | 17.93 | 19.79 | + | + | Kriege & Mutzel (2012) |
| MSRC_21C | Vision | 209 | 20 | 40.28 | 96.6 | + | - | Neumann et al. (2016) |
| MSRC_9 | Vision | 221 | 8 | 40.58 | 97.94 | + | - | Neumann et al. (2016) |
| Cuneiform | Molecular | 267 | 30 | 21.27 | 44.8 | + | + | Kriege et al. (2018a) |
| SYNTHETIC | Synthetic | 300 | 2 | 100 | 196 | - | - | Feragen et al. (2013) |
| COX2_MD | Molecular | 303 | 2 | 26.28 | 335.12 | + | + | Kriege & Mutzel (2012) |
| BZR_MD | Molecular | 306 | 2 | 21.3 | 225.06 | + | + | Kriege & Mutzel (2012) |
| PTC_MM | Molecular | 336 | 2 | 13.97 | 14.32 | + | + | Kriege & Mutzel (2012) |
| PTC_MR | Molecular | 344 | 2 | 14.29 | 14.69 | + | + | Kriege & Mutzel (2012) |
| PTC_FM | Molecular | 349 | 2 | 14.11 | 14.48 | + | + | Kriege & Mutzel (2012) |
| PTC_FR | Molecular | 351 | 2 | 14.56 | 15 | + | + | Kriege & Mutzel (2012) |
| DHFR_MD | Molecular | 393 | 2 | 23.87 | 283.01 | + | + | Kriege & Mutzel (2012) |
| Synthie | Synthetic | 400 | 4 | 95 | 172.93 | - | - | Morris et al. (2016) |
| BZR | Molecular | 405 | 2 | 35.75 | 38.36 | + | - | Sutherland et al. (2003) |
| ER_MD | Molecular | 446 | 2 | 21.33 | 234.85 | + | + | Kriege & Mutzel (2012) |
| COX2 | Molecular | 467 | 2 | 41.22 | 43.45 | + | - | Sutherland et al. (2003) |
| DHFR | Molecular | 467 | 2 | 42.43 | 44.54 | + | - | Sutherland et al. (2003) |
| MSRC_21 | Vision | 563 | 20 | 77.52 | 198.32 | + | - | Neumann et al. (2016) |
| ENZYMES | Molecular | 600 | 6 | 32.63 | 62.14 | + | - | Borgwardt et al. (2005) |
| IMDB-BINARY | Social | 1000 | 2 | 19.77 | 96.53 | - | - | Yanardag & Vishwanathan (2015) |
| PROTEINS | Molecular | 1113 | 2 | 39.06 | 72.82 | + | - | Borgwardt et al. (2005) |
| DD | Molecular | 1178 | 2 | 284.32 | 715.66 | + | - | Shervashidze et al. (2011a) |
| IMDB-MULTI | Social | 1500 | 3 | 13 | 65.94 | - | - | Yanardag & Vishwanathan (2015) |
| AIDS | Molecular | 2000 | 2 | 15.69 | 16.2 | + | + | Riesen & Bunke (2008) |
| REDDIT-BINARY | Social | 2000 | 2 | 429.63 | 497.75 | - | - | Yanardag & Vishwanathan (2015) |
| Letter-high | Molecular | 2250 | 15 | 4.67 | 4.5 | - | - | Riesen & Bunke (2008) |
| Letter-low | Molecular | 2250 | 15 | 4.68 | 3.13 | - | - | Riesen & Bunke (2008) |
| Letter-med | Molecular | 2250 | 15 | 4.67 | 4.5 | - | - | Riesen & Bunke (2008) |
| Fingerprint | Molecular | 2800 | 4 | 5.42 | 4.42 | - | - | Riesen & Bunke (2008) |
| COIL-DEL | Molecular | 3900 | 100 | 21.54 | 54.24 | - | + | Riesen & Bunke (2008) |
| COIL-RAG | Molecular | 3900 | 100 | 3.01 | 3.02 | - | - | Riesen & Bunke (2008) |
| NCI1 | Molecular | 4110 | 2 | 29.87 | 32.3 | + | - | Shervashidze et al. (2011a) |
| NCI109 | Molecular | 4127 | 2 | 29.68 | 32.13 | + | - | Shervashidze et al. (2011a) |
| FRANKENSTEIN | Molecular | 4337 | 2 | 16.9 | 17.88 | - | - | Orsini et al. (2015) |
| Mutagenicity | Molecular | 4337 | 2 | 30.32 | 30.77 | + | + | Riesen & Bunke (2008) |
| REDDIT-MULTI-5K | Social | 4999 | 5 | 508.52 | 594.87 | - | - | Yanardag & Vishwanathan (2015) |
| COLLAB | Social | 5000 | 3 | 74.49 | 2457.78 | - | - | Yanardag & Vishwanathan (2015) |
| Tox21_ARE | Molecular | 7167 | 2 | 16.28 | 16.52 | + | + | Challenge (2014) |
| Tox21_aromatase | Molecular | 7226 | 2 | 17.5 | 17.79 | + | + | Challenge (2014) |
| Tox21_MMP | Molecular | 7320 | 2 | 17.49 | 17.83 | + | + | Challenge (2014) |
| Tox21_ER | Molecular | 7697 | 2 | 17.58 | 17.94 | + | + | Challenge (2014) |
| Tox21_HSE | Molecular | 8150 | 2 | 16.72 | 17.04 | + | + | Challenge (2014) |
| Tox21_AHR | Molecular | 8169 | 2 | 18.09 | 18.5 | + | + | Challenge (2014) |
| Tox21_PPAR-gamma | Molecular | 8184 | 2 | 17.23 | 17.55 | + | + | Challenge (2014) |
| Tox21_AR-LBD | Molecular | 8599 | 2 | 17.77 | 18.16 | + | + | Challenge (2014) |
| Tox21_p53 | Molecular | 8634 | 2 | 17.79 | 18.19 | + | + | Challenge (2014) |
| Tox21_ER_LBD | Molecular | 8753 | 2 | 18.06 | 18.47 | + | + | Challenge (2014) |
| Tox21_ATAD5 | Molecular | 9091 | 2 | 17.89 | 18.3 | + | + | Challenge (2014) |
| Tox21_AR | Molecular | 9362 | 2 | 18.39 | 18.84 | + | + | Challenge (2014) |
| REDDIT-MULTI-12K | Social | 11929 | 11 | 391.41 | 456.89 | - | - | Yanardag & Vishwanathan (2015) |

# B    ISOMORPHISM METRICS FOR ALL DATA SETS

Table 6: Isomorphic metrics for all data sets. Sorting is based on the proportion of isomorphic graphs $I\%$. Num. orbits is the number of non-trivial orbits. $IP\%$ is the proportion of isomorphic pairs of graphs, Mismatched $\%$ is the proportion of mismatched labels.

| data set | Size, $N$ | Num. orbits | Iso. graphs, $I$ | $I\%$ | $IP\%$ | Mismatched $\%$ |
|---|---|---|---|---|---|---|
| SYNTHETIC | 300 | 2 | 300 | 100 | 100 | 100 |
| Cuneiform | 267 | 8 | 267 | 100 | 20.46 | 100 |
| Letter-low | 2250 | 32 | 2245 | 99.78 | 8.72 | 96.22 |
| DHFR_MD | 393 | 25 | 392 | 99.75 | 6.87 | 94.91 |
| COIL-RAG | 3900 | 20 | 3890 | 99.74 | 25.22 | 99.31 |
| COX2_MD | 303 | 13 | 301 | 99.34 | 11.83 | 98.68 |
| ER_MD | 446 | 31 | 442 | 99.1 | 5.57 | 82.74 |
| Fingerprint | 2800 | 69 | 2774 | 99.07 | 16.86 | 89.29 |
| BZR_MD | 306 | 22 | 303 | 99.02 | 7.16 | 95.75 |
| Letter-med | 2250 | 39 | 2226 | 98.93 | 8.05 | 92.93 |
| Letter-high | 2250 | 94 | 2200 | 97.78 | 3.67 | 95.91 |
| IMDB-MULTI | 1500 | 100 | 1212 | 80.8 | 6.39 | 74.67 |
| Tox21_ATAD5 | 9091 | 1461 | 6167 | 67.84 | 0.09 | 9.15 |
| Tox21_PPAR-gamma | 8184 | 1265 | 5513 | 67.36 | 0.1 | 7.77 |
| Tox21_AR | 9362 | 1519 | 6295 | 67.24 | 0.08 | 8 |
| Tox21_p53 | 8634 | 1345 | 5800 | 67.18 | 0.09 | 11.28 |
| Tox21_AR-LBD | 8599 | 1354 | 5766 | 67.05 | 0.09 | 6.88 |
| Tox21_MMP | 7320 | 1138 | 4875 | 66.6 | 0.1 | 18.76 |
| Tox21_HSE | 8150 | 1218 | 5425 | 66.56 | 0.1 | 18.02 |
| Tox21_ER_LBD | 8753 | 1375 | 5791 | 66.16 | 0.09 | 12.41 |
| Tox21_ER | 7697 | 1203 | 5078 | 65.97 | 0.09 | 27.32 |
| Tox21_AHR | 8169 | 1299 | 5377 | 65.82 | 0.09 | 15.61 |
| Tox21_aromatase | 7226 | 1084 | 4727 | 65.42 | 0.1 | 5.07 |
| Tox21_ARE | 7167 | 1047 | 4682 | 65.33 | 0.11 | 26.45 |
| AIDS | 2000 | 371 | 1259 | 62.95 | 0.13 | 0.35 |
| COX2 | 467 | 76 | 283 | 60.6 | 0.6 | 20.56 |
| IMDB-BINARY | 1000 | 117 | 579 | 57.9 | 0.67 | 31.8 |
| FRANKENSTEIN | 4337 | 574 | 2230 | 51.42 | 0.09 | 30.87 |
| MUTAG | 188 | 31 | 79 | 42.02 | 0.49 | 6.91 |
| BZR | 405 | 43 | 165 | 40.74 | 0.6 | 8.89 |
| PTC_MM | 336 | 42 | 132 | 39.29 | 0.46 | 23.21 |
| PTC_MR | 344 | 40 | 125 | 36.34 | 0.41 | 25 |
| PTC_FM | 349 | 39 | 124 | 35.53 | 0.39 | 23.5 |
| DHFR | 756 | 89 | 250 | 33.07 | 0.14 | 9.13 |
| PTC_FR | 351 | 36 | 116 | 33.05 | 0.37 | 20.51 |
| Mutagenicity | 4337 | 397 | 1274 | 29.38 | 0.03 | 13.1 |
| COLLAB | 5000 | 158 | 1077 | 21.54 | 0.11 | 6.68 |
| COIL-DEL | 3900 | 155 | 796 | 20.41 | 0.06 | 18.56 |
| PROTEINS | 1113 | 35 | 151 | 13.57 | 0.1 | 9.07 |
| NCI1 | 4110 | 225 | 523 | 12.73 | 0.01 | 1.7 |
| NCI109 | 4127 | 222 | 519 | 12.58 | 0.01 | 1.7 |
| ENZYMES | 600 | 6 | 10 | 1.67 | 0 | 0 |
| REDDIT-BINARY | 2000 | 3 | 4 | 0.2 | 0 | 0 |
| REDDIT-MULTI-12K | 11929 | 8 | 17 | 0.14 | 0 | 0.04 |
| FIRSTMM_DB | 41 | 1 | 0 | 0 | 0 | 0 |
| OHSU | 79 | 1 | 0 | 0 | 0 | 0 |
| KKI | 83 | 1 | 0 | 0 | 0 | 0 |
| Peking_1 | 85 | 1 | 0 | 0 | 0 | 0 |
| MSRC_21C | 209 | 1 | 0 | 0 | 0 | 0 |
| MSRC_9 | 221 | 1 | 0 | 0 | 0 | 0 |
| Synthie | 400 | 1 | 0 | 0 | 0 | 0 |
| MSRC_21 | 563 | 1 | 0 | 0 | 0 | 0 |
| DD | 1178 | 1 | 0 | 0 | 0 | 0 |
| REDDIT-MULTI-5K | 4999 | 1 | 0 | 0 | 0 | 0 |

## C ORBIT SIZE DISTRIBUTION FOR ALL DATA SETS

In the plots 3, 4, 5 the sizes of orbits are presented for each data set. Empty plots correspond to data sets with no isomorphic graphs. Plots with just wo-labels correspond to cases when there are no node labels available for the graphs in a data set.

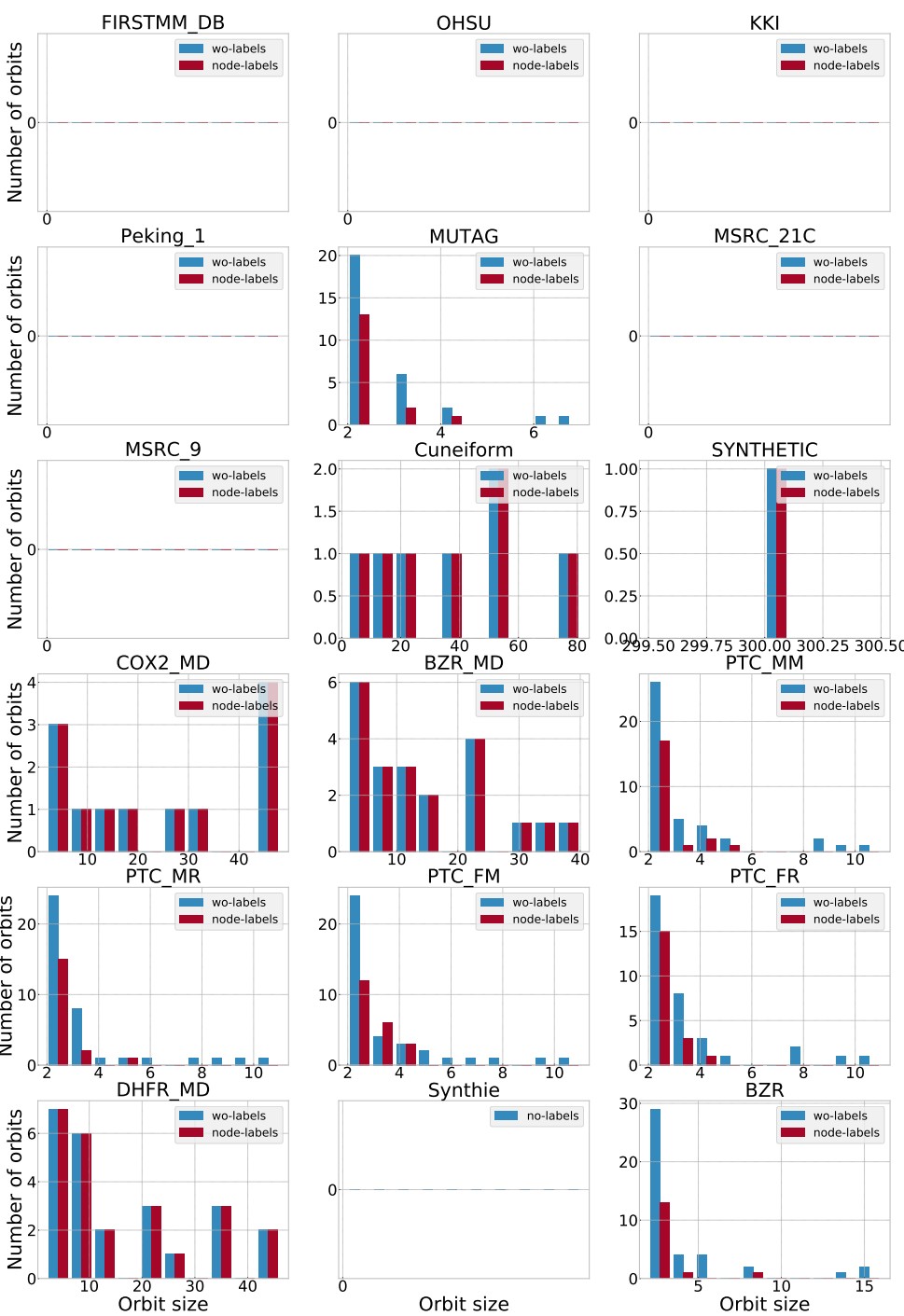

Figure 3: Distribution of orbits sizes. wo-labels correspond to isomorphism without considering the labels. node-labels correspond to isomorphism that considers node labels. Part-1.

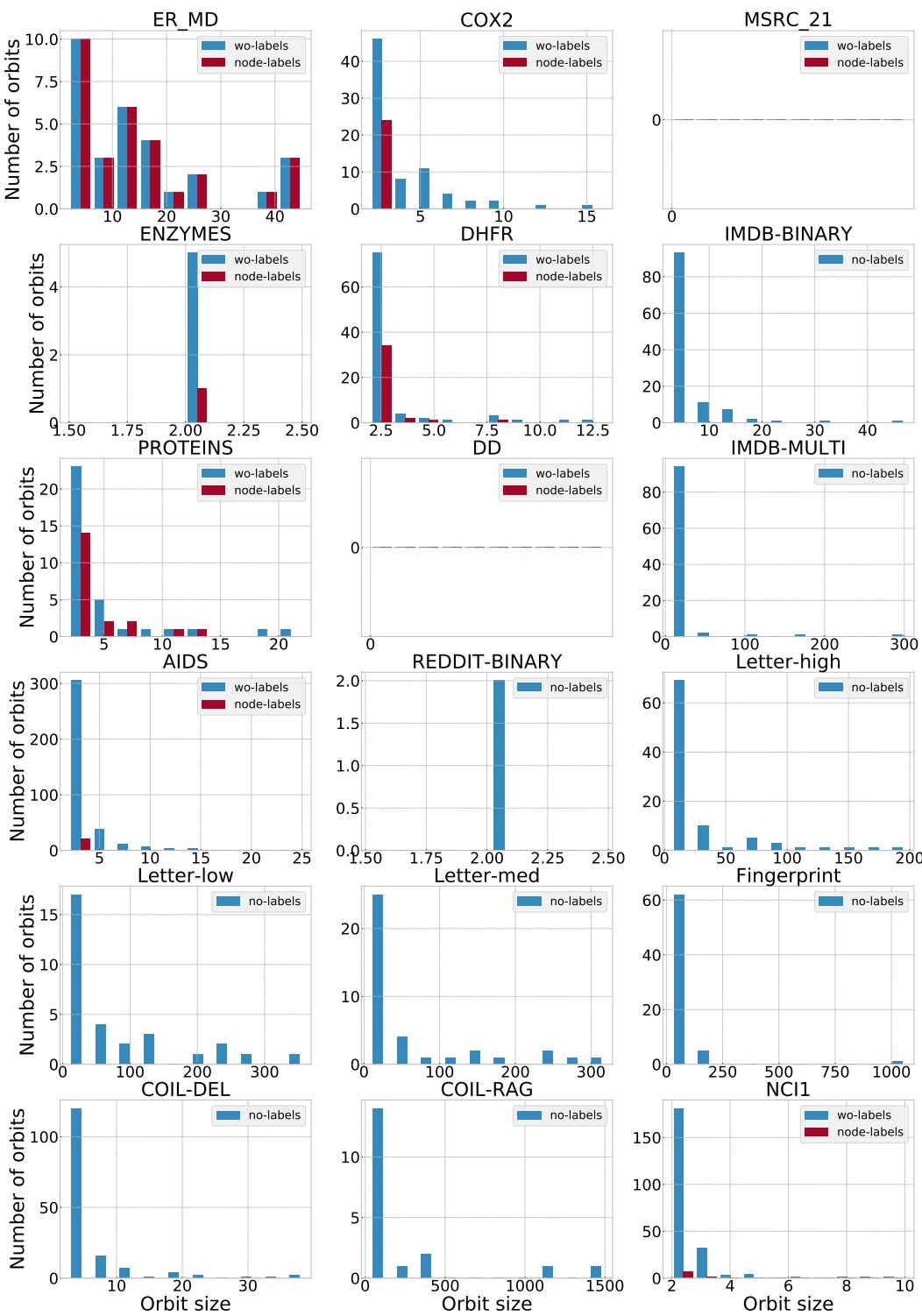

Figure 4: Distribution of orbits sizes. Part-2.

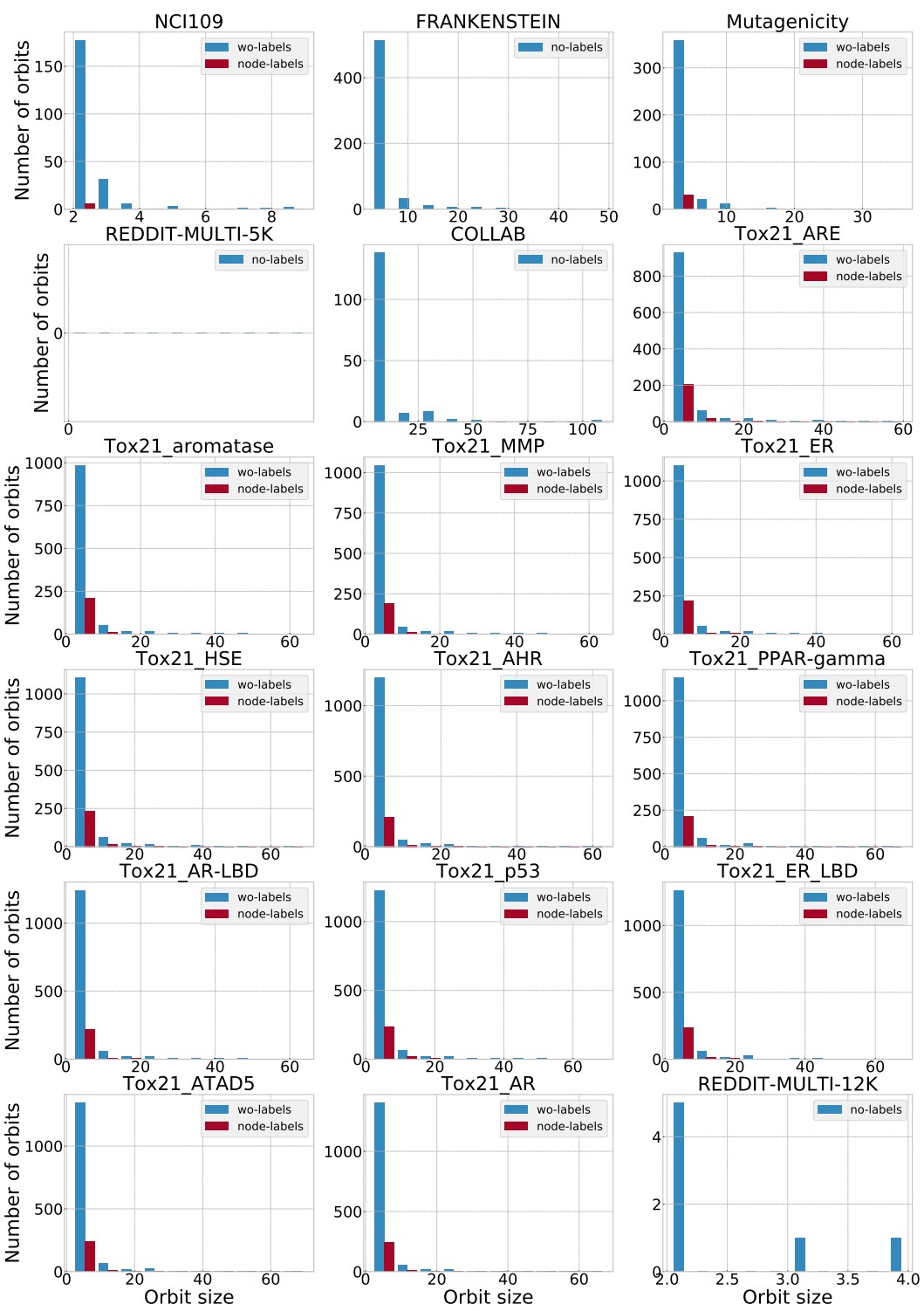

Figure 5: Distribution of orbits sizes. Part-3.

# D    ISOMORPHISM METRICS FOR DATA SETS WITH NODE LABELS

Table 7: Isomorphic metrics for data sets with node labels. Sorting is based on the proportion of isomorphic graphs $I\%$. Num. orbits is the number of non-trivial orbits. $IP\%$ is the proportion of isomorphic pairs of graphs, Mismatched $\%$ is the proportion of mismatched labels. Table does not include data sets with no node labels.

| data set | Size, $N$ | Num. orbits | Iso. graphs, $I$ | $I\%$ | $IP\%$ | Mismatched % |
|---|---|---|---|---|---|---|
| SYNTHETIC | 300 | 2 | 300 | 100 | 100 | 100 |
| Cuneiform | 267 | 8 | 267 | 100 | 20.46 | 100 |
| DHFR_MD | 393 | 25 | 392 | 99.75 | 6.87 | 94.91 |
| COX2_MD | 303 | 13 | 301 | 99.34 | 11.83 | 98.68 |
| ER_MD | 446 | 31 | 442 | 99.1 | 5.57 | 82.74 |
| BZR_MD | 306 | 22 | 303 | 99.02 | 7.16 | 95.75 |
| MUTAG | 188 | 17 | 36 | 19.15 | 0.14 | 0 |
| PTC_FM | 349 | 22 | 54 | 15.47 | 0.08 | 10.89 |
| PTC_MM | 336 | 22 | 50 | 14.88 | 0.07 | 7.74 |
| DHFR | 756 | 39 | 98 | 12.96 | 0.04 | 3.97 |
| PTC_FR | 351 | 20 | 43 | 12.25 | 0.05 | 6.27 |
| PTC_MR | 344 | 19 | 41 | 11.92 | 0.05 | 6.4 |
| Tox21_ARE | 7167 | 228 | 820 | 11.44 | 0.01 | 3.91 |
| Tox21_HSE | 8150 | 250 | 919 | 11.28 | 0.01 | 2.25 |
| Tox21_aromatase | 7226 | 223 | 805 | 11.14 | 0.01 | 0.77 |
| Tox21_p53 | 8634 | 258 | 946 | 10.96 | 0.01 | 1.83 |
| Tox21_ER | 7697 | 231 | 840 | 10.91 | 0.01 | 2.4 |
| COX2 | 467 | 25 | 50 | 10.71 | 0.03 | 1.07 |
| Tox21_PPAR-gamma | 8184 | 227 | 869 | 10.62 | 0.01 | 0.76 |
| Tox21_MMP | 7320 | 204 | 770 | 10.52 | 0.01 | 2.19 |
| Tox21_ER_LBD | 8753 | 255 | 913 | 10.43 | 0.01 | 1.28 |
| Tox21_AR | 9362 | 265 | 965 | 10.31 | 0.01 | 0.68 |
| Tox21_ATAD5 | 9091 | 255 | 924 | 10.16 | 0.01 | 1.04 |
| Tox21_AR-LBD | 8599 | 238 | 871 | 10.13 | 0.01 | 0.7 |
| BZR | 405 | 16 | 40 | 9.88 | 0.06 | 0.99 |
| Tox21_AHR | 8169 | 224 | 795 | 9.73 | 0.01 | 2.07 |
| PROTEINS | 1113 | 21 | 74 | 6.65 | 0.03 | 2.61 |
| AIDS | 2000 | 22 | 54 | 2.7 | 0 | 0 |
| Mutagenicity | 4337 | 31 | 75 | 1.73 | 0 | 0.92 |
| NCI1 | 4110 | 9 | 17 | 0.41 | 0 | 0.05 |
| ENZYMES | 600 | 2 | 2 | 0.33 | 0 | 0 |
| NCI109 | 4127 | 7 | 12 | 0.29 | 0 | 0.05 |
| FIRSTMM_DB | 41 | 1 | 0 | 0 | 0 | 0 |
| OHSU | 79 | 1 | 0 | 0 | 0 | 0 |
| KKI | 83 | 1 | 0 | 0 | 0 | 0 |
| Peking_1 | 85 | 1 | 0 | 0 | 0 | 0 |
| MSRC_21C | 209 | 1 | 0 | 0 | 0 | 0 |
| MSRC_9 | 221 | 1 | 0 | 0 | 0 | 0 |
| MSRC_21 | 563 | 1 | 0 | 0 | 0 | 0 |
| DD | 1178 | 1 | 0 | 0 | 0 | 0 |

## E    PROOF OF CLAIM 7.1

*Proof.* From the equation 3 we have:

$$\text{acc}(Y_{test}) = \frac{|Y_{new}|}{|Y_{test}|}\text{acc}(Y_{new}) + \frac{|Y_{iso}|}{|Y_{test}|}\text{acc}(Y_{iso}) > \text{acc}(Y_{iso}) \Rightarrow$$

$$\frac{|Y_{new}|}{|Y_{test}|}\text{acc}(Y_{new}) > (1 - \frac{|Y_{iso}|}{|Y_{test}|})\text{acc}(Y_{iso}) \Rightarrow$$

$$\text{acc}(Y_{iso}) > \text{acc}(Y_{new})$$

$\square$

## F    PROOF OF CLAIM 7.2

*Proof.* From the definition of the peering model we have:

$$\text{acc}_{\mathcal{M}}(Y_{new}) = \text{acc}_{\widehat{\mathcal{M}}}(Y_{new})$$

$$\text{acc}_{\mathcal{M}}(Y_{iso}) \leq \text{acc}_{\widehat{\mathcal{M}}}(Y_{iso}) = 1$$

Substituting these into the equation 3 we have:

$$\text{acc}_{\mathcal{M}}(Y_{test}) \leq \text{acc}_{\widehat{\mathcal{M}}}(Y_{test}).$$

$\square$

## G    EXPERIMENTATION DETAILS

NN model is from Xu et al. (2018) and evaluate it on the data sets from PyTorch-Geometric (Fey & Lenssen, 2019). For each data set we perform 10-fold cross-validation such that each fold is evaluated on 10% of hold-out instances $Y_{test}$. For each fold we train the model for 350 epochs selecting the final model with the best performance on the validation set (20% from hold-out trained split) across all epochs. Additionally we found that for small data set performance during the first epochs can be unstable on the validation set and thus we select our model only after the first 50 epochs. The final model is evaluated on the test instances and corresponds to **NN** in the experiments. Peering models **NN-PH** and **NN-P** are obtained from **NN** by replicating the target labels for homogeneous $Y_{iso}$ and non-homogeneous $Y_{iso}$ respectively. Weisfeiler-Lehman and Vertex histogram kernels are taken from the code[2] of Sugiyama & Borgwardt (2015). We selected the height of subtree $h = 5$ for WL kernel. We train an SVM model selecting $C$ parameter from the range [0.001, 0.01, 0.1, 1, 10].

---

[2]`https://github.com/BorgwardtLab/graph-kernels`

# H  NEW DATA SETS

Table 8: Statistics for new clean data sets. Retention is the proportion of graphs remaining after cleaning procedure. Min. Class and Max. Class are minimum and maximum number of graphs in a class. Sorted by retention.

| Data set | Size | Retention, % | Avg. Nodes | Avg. Edges | Classes | Min. Class | Max. Class |
|---|---|---|---|---|---|---|---|
| SYNTHETIC | 0 | 0 | 0 | 0 | 0 | 0 | 0 |
| Cuneiform | 0 | 0 | 0 | 0 | 0 | 0 | 0 |
| COIL-RAG | 13 | 0.33 | 5.77 | 9.62 | 7 | 1 | 5 |
| Letter-low | 12 | 0.53 | 7.17 | 5.42 | 5 | 1 | 3 |
| COX2_MD | 3 | 0.99 | 30.33 | 467 | 2 | 1 | 2 |
| DHFR_MD | 4 | 1.02 | 25.25 | 366.25 | 2 | 1 | 3 |
| Letter-med | 29 | 1.29 | 6.83 | 5.66 | 8 | 1 | 10 |
| Fingerprint | 51 | 1.82 | 14.45 | 13.12 | 6 | 1 | 40 |
| BZR_MD | 6 | 1.96 | 14.17 | 130.17 | 2 | 1 | 5 |
| Letter-high | 60 | 2.67 | 7.03 | 7.23 | 7 | 1 | 32 |
| ER_MD | 14 | 3.14 | 18.07 | 227.36 | 2 | 1 | 13 |
| IMDB-MULTI | 321 | 21.4 | 22.35 | 249.46 | 3 | 85 | 144 |
| Tox21_ARE | 3302 | 46.07 | 20.96 | 21.86 | 2 | 602 | 2700 |
| Tox21_ER | 3560 | 46.25 | 22.24 | 23.29 | 2 | 419 | 3141 |
| Tox21_MMP | 3405 | 46.52 | 22.4 | 23.43 | 2 | 618 | 2787 |
| Tox21_HSE | 3814 | 46.8 | 21.29 | 22.28 | 2 | 207 | 3607 |
| Tox21_p53 | 4088 | 47.35 | 22.62 | 23.72 | 2 | 291 | 3797 |
| Tox21_PPAR-gamma | 3877 | 47.37 | 21.91 | 22.91 | 2 | 120 | 3757 |
| Tox21_ATAD5 | 4312 | 47.43 | 22.52 | 23.61 | 2 | 168 | 4144 |
| Tox21_AR-LBD | 4134 | 48.08 | 22.41 | 23.48 | 2 | 150 | 3984 |
| Tox21_AR | 4506 | 48.13 | 22.93 | 24.05 | 2 | 189 | 4317 |
| Tox21_AHR | 3935 | 48.17 | 22.88 | 23.98 | 2 | 490 | 3445 |
| Tox21_ER_LBD | 4224 | 48.26 | 22.69 | 23.8 | 2 | 193 | 4031 |
| Tox21_aromatase | 3524 | 48.77 | 22.24 | 23.22 | 2 | 234 | 3290 |
| IMDB-BINARY | 493 | 49.3 | 24.08 | 221.96 | 2 | 232 | 261 |
| COX2 | 237 | 50.75 | 42.14 | 44.43 | 2 | 68 | 169 |
| AIDS | 1110 | 55.5 | 18.22 | 19.1 | 2 | 310 | 800 |
| FRANKENSTEIN | 2448 | 56.44 | 20.78 | 22.35 | 2 | 1020 | 1428 |
| PTC_MM | 226 | 67.26 | 17.04 | 17.74 | 2 | 77 | 149 |
| BZR | 276 | 68.15 | 36.25 | 38.81 | 2 | 72 | 204 |
| PTC_MR | 235 | 68.31 | 17.23 | 17.97 | 2 | 96 | 139 |
| PTC_FM | 242 | 69.34 | 16.96 | 17.66 | 2 | 85 | 157 |
| MUTAG | 135 | 71.81 | 18.85 | 20.84 | 2 | 42 | 93 |
| PTC_FR | 253 | 72.08 | 17.11 | 17.86 | 2 | 86 | 167 |
| DHFR | 578 | 76.46 | 43.37 | 45.53 | 2 | 205 | 373 |
| Mutagenicity | 3335 | 76.9 | 32.96 | 34.16 | 2 | 1484 | 1851 |
| COIL-DEL | 3133 | 80.33 | 25.05 | 64.26 | 98 | 1 | 39 |
| COLLAB | 4064 | 81.28 | 76.94 | 4667.92 | 3 | 770 | 2289 |
| PROTEINS | 975 | 87.6 | 43.41 | 81.04 | 2 | 343 | 632 |
| NCI1 | 3785 | 92.09 | 29.84 | 32.37 | 2 | 1781 | 2004 |
| NCI109 | 3801 | 92.1 | 29.66 | 32.22 | 2 | 1801 | 2000 |
| ENZYMES | 595 | 99.17 | 32.48 | 62.17 | 6 | 98 | 100 |
| REDDIT-BINARY | 1998 | 99.9 | 430.04 | 996.48 | 2 | 998 | 1000 |
| REDDIT-MULTI-12K | 11917 | 99.9 | 391.79 | 914.68 | 11 | 513 | 2586 |
| FIRSTMM_DB | 41 | 100 | 1377.27 | 3073.93 | 11 | 2 | 6 |
| OHSU | 79 | 100 | 82.01 | 199.66 | 2 | 35 | 44 |
| KKI | 83 | 100 | 26.96 | 48.42 | 2 | 37 | 46 |
| Peking_1 | 85 | 100 | 39.31 | 77.35 | 2 | 36 | 49 |
| MSRC_21C | 209 | 100 | 40.28 | 96.6 | 17 | 1 | 29 |
| MSRC_9 | 221 | 100 | 40.58 | 97.94 | 8 | 19 | 30 |
| Synthie | 400 | 100 | 91.6 | 202.18 | 4 | 90 | 110 |
| MSRC_21 | 563 | 100 | 77.52 | 198.32 | 20 | 10 | 34 |
| DD | 1178 | 100 | 284.32 | 715.66 | 2 | 487 | 691 |
| REDDIT-MULTI-5K | 4999 | 100 | 508.51 | 1189.75 | 5 | 999 | 1000 |

# I    PROOF OF THEOREM 6.1

*Proof.* Let us prove Theorem 6.1. We denote by $\mathcal{L} = \{(x, y) \rightarrow L(f(x), y) : f \in \mathcal{F}\}$ a composite loss class. For any $L \in \mathcal{L}$ we get that

$$\mathbb{E}_{\mathbb{P}}L - \mathbb{E}_{\mathcal{D}}uL = \mathbb{E}_{\mathbb{P}}L - \mathbb{E}_{\mathbb{P}}uL + \mathbb{E}_{\mathbb{P}}uL - \mathbb{E}_{\mathcal{D}}uL \leq$$
$$\leq \mathbb{E}_{\mathbb{P}}|(1-u)L| + (\mathbb{E}_{\mathbb{P}}uL - \mathbb{E}_{\mathcal{D}}uL). \tag{5}$$

Since any $L \in \mathcal{L}$ is bounded from above by 1 for the first term in equation 5 we obtain

$$\mathbb{E}_{\mathbb{P}}|(1-u)L| \leq \mathbb{E}_{\mathbb{P}}g_+(w)\mathbb{I}_{\{y=+1\}} + \mathbb{E}_{\mathbb{P}}g_-(w)\mathbb{I}_{\{y=-1\}} = g_+(w)\pi + g_-(w)(1-\pi). \tag{6}$$

Thanks to McDiarmid'd concentration inequality Mohri et al. (2012), applied to the function class $\mathcal{L}_u = \{uL : L \in \mathcal{L}\}$, with probability $1 - \delta$, $\delta > 0$ for $\mathcal{D} \sim \mathbb{P}^N$ we get the upper bound on the excess risk

$$\sup_{L \in \mathcal{L}}(\mathbb{E}_{\mathbb{P}}uL - \mathbb{E}_{\mathcal{D}}uL) \leq 2\mathcal{R}_N(\mathcal{L}_u) + \max[(1 + g_+(w)), (1 + g_-(w))]\sqrt{(\log \delta^{-1})/(2N)} \leq$$
$$\leq 2\mathcal{R}_N(\mathcal{L}_u) + (2 + g_+(w) + g_-(w))\sqrt{(\log \delta^{-1})/(2N)}. \tag{7}$$

Let us find a relation between $\mathcal{R}_N(\mathcal{L}_u)$ and $\mathcal{R}_N(\mathcal{L})$. Here we denote by $z_i$ a pair $z_i = (x_i, y_i)$. By the definition (see Mohri et al. (2012)) the empirical Rademacher complexity

$$\hat{\mathcal{R}}_{\mathcal{D}}(\mathcal{L}_u) = \frac{1}{N}\mathbb{E}_{\sigma}\sup_{L \in \mathcal{L}_u}\sum_{i=1}^{N}\sigma_i u(z_i)L(z_i) \leq$$

$$\leq \frac{1}{N}\mathbb{E}_{\sigma}\sup_{L \in \mathcal{L}_u}\sum_{i=1}^{N}\sigma_i L(z_i) + \frac{g_+(w)}{N}\mathbb{E}_{\sigma}\sup_{L \in \mathcal{L}_u}\sum_{i:y_i=+1}\sigma_i L(z_i) + \frac{g_-(w)}{N}\mathbb{E}_{\sigma}\sup_{L \in \mathcal{L}_u}\sum_{i:y_i=-1}\sigma_i L(z_i) \leq$$

$$\leq \hat{\mathcal{R}}_{\mathcal{D}}(\mathcal{L}) + \frac{g_+(w)}{N}\mathbb{E}_{\sigma}\sup_{L \in \mathcal{L}_u}\sum_{i:y_i=+1}\sigma_i L(z_i) + \frac{g_-(w)}{N}\mathbb{E}_{\sigma}\sup_{L \in \mathcal{L}_u}\sum_{i:y_i=-1}\sigma_i L(z_i).$$

Since we use the zero-one loss, then

$$\mathbb{E}_{\sigma}\sup_{L \in \mathcal{L}_u}\sum_{i:y_i=+1}\sigma_i L(z_i) \leq \#\{i : y_i = +1\}, \quad \mathbb{E}_{\sigma}\sup_{L \in \mathcal{L}_u}\sum_{i:y_i=-1}\sigma_i L(z_i) \leq \#\{i : y_i = -1\}.$$

The Rademacher complexity

$$\mathcal{R}_N(\mathcal{L}_u) = \mathbb{E}_{\mathcal{D} \sim \mathbb{P}^N}\hat{\mathcal{R}}_{\mathcal{D}}(\mathcal{L}_u) \leq \mathbb{E}_{\mathcal{D} \sim \mathbb{P}^N}\left[\hat{\mathcal{R}}_{\mathcal{D}}(\mathcal{L}) + \frac{g_+(w)}{N}\#\{i : y_i = +1\}+\right.$$
$$\left. + \frac{g_-(w)}{N}\#\{i : y_i = -1\}\right] = \mathcal{R}_N(\mathcal{L}) + g_+(w)\pi + g_-(w)(1-\pi). \tag{8}$$

Using the fact that $\mathcal{R}_N(\mathcal{L}) = \frac{1}{2}\mathcal{R}_N(\mathcal{F})$, substituting inequalities 6, 7 and 8 into equation 5, we get that

$$\sup_{f \in \mathcal{F}}\left(\mathbb{E}_{\mathbb{P}}l(f(x), y) - \mathbb{E}_{\mathcal{D}}u(x, y)l(f(x), y)\right) \leq 3\left(g_+(w)\pi + g_-(w)(1-\pi)\right)+$$
$$+ \mathcal{R}_N(\mathcal{F}) + (2 + g_+(w) + g_-(w))\sqrt{(\log \delta^{-1})/(2N)}.$$

$\square$

