# OpenReview forum: "Understanding Isomorphism Bias in Graph Data Sets "
_ICLR.cc/2020/Conference — Reject_

### Official Review · AnonReviewer3 · 2019-10-21
**Official Blind Review #3**

**Rating:** 6

**Review:**

The authors discuss here the problem of isomorphism bias in graph dataset, i.e. the overfitting effect in learning networks whenever graph isomorphism features are incorporated within the model. This is a bias which jeopardises the validity and the reproducibility of several studies, and it is theoretically analogous to data leakage effects.
The authors fairly discuss the problem in the introduction, with a good coverage of the related literature; the background theory is reasonably discussed, although is not very deep. The experimental part is extensive and well described, and it shows the overfitting effect very clearly. However, the novelty of the work is limited, and also the proposed solutions cannot be claimed as superior to other approaches, due to the small improvement in accuracy.

**Experience Assessment:**

I have published one or two papers in this area.

**Review Assessment: Checking Correctness Of Derivations And Theory:**

I carefully checked the derivations and theory.

**Review Assessment: Checking Correctness Of Experiments:**

I carefully checked the experiments.

**Review Assessment: Thoroughness In Paper Reading:**

I read the paper thoroughly.

---

> ### Author Response · Authors · 2019-11-06
> **Thank you for your review**
>
> We appreciate your positive feedback.
>
> Indeed, our goal was to provide a thorough analysis of popular graph data sets and to define new steps according to our analysis. Unlike other popular papers in this area, we do not propose new classification algorithms, but rather take a look at another important aspect of classification problem, namely the data. This is a novel way to look at this problem which provides new insights into the performance of current classification models. Additionally, the results can have a large impact on the domain of graph classification as a whole, because it establishes a fair comparison between future models, which in turn can lead to the developments of new graph representation learning methods.

---

### Official Review · AnonReviewer5 · 2019-11-01
**Official Blind Review #5**

**Rating:** 1

**Review:**

The paper presents three contributions: (a) the observation that there’s train-to-test leakage in many graph classification datasets (under isomorphism equivalence), (b) what appears to be a theoretically motivated way of improving scores on such datasets, by focusing on solving the examples that are isomorphic with training instances, and (c) a recommendation to remove such leakage from test sets. I don’t think the paper meets the ICLR bar. While (a) is very interesting, and an important contribution, (b) and (c) are contradictory. The recommendation (c) is a bit of a no-brainer, and Property 6.1 and Theorem 6.1, providing the substance of (b), are near-trivial.

Missing reference: Bordes et al. (2013) and Toutanova et al. (2015) show there’s train-to-test leakage (under isomorphism equivalence) in the FB15K dataset.

**Experience Assessment:**

I have read many papers in this area.

**Review Assessment: Checking Correctness Of Derivations And Theory:**

N/A

**Review Assessment: Checking Correctness Of Experiments:**

I did not assess the experiments.

**Review Assessment: Thoroughness In Paper Reading:**

I made a quick assessment of this paper.

---

> ### Author Response · Authors · 2019-11-06
> **We appreciate your feedback**
>
> Our main contribution is the thorough analysis of current graph classification data sets and a proposal of a new framework for proper validation of the models. Please note that running pairwise isomorphism test on 54 data sets requires diligence and time. For example, discovering all isomorphic pairs for the largest data set requires checking more than 71 million graph pairs. As graph isomorphism does not have a general polynomial-time algorithm, we had to write very efficient code and even with that, it takes hours to get all statistics for all data sets. One of the important deliveries is that anyone can run an efficient graph isomorphism test for any graph data set.
>
> Another contribution is that we show that the problem of having isomorphic graphs is two-fold. First, a model can exploit train-to-test leakage as we show in Table 4. This is also backed-up by proposition and theorem statements. Second is that even if a model does not exploit a leak, final validation metrics can be biased by isomorphic instances and hence validation does not test the generalization ability of a model.
>
> To the best of our judgment, we are also the first to explain the presence of so many isomorphic graphs in the commonly used data sets. Often it is a problem of discarding meta-information from consideration as is shown in Table 7. But not only: other factors such as sizes of graphs or the origin of the data sets may also explain this phenomenon. This is a useful insight for graph practitioners to consider when designing new data sets.
>
> While the paper is written in an intuitive manner, we hope that it does not underestimate the contributions that it makes. The results are computationally-challenging which saves a lot of time for future research in this area. Graph classification problem has been dominant for representation learning of graphs, leading to advancements in Graph Neural Networks research, and thus clean data sets are required for learning good graph embeddings. We also note that many of the data sets come from the Biological application, one of the ICLR main applications (as denoted in the main website) and some results of this paper provide new insights into this application (e.g. noisy target labels for many graphs; necessity to use node/edge attributes; etc.). Besides, the results about current data sets may have a large impact on the whole area of graph classification as it provides new insights on the performance of current and future classification models and makes sure the proposed validation framework is correct.
>
> We hope that this response is useful and you can adjust your rating to our submission. If you have any further concerns, we would be happy to discuss them.

---

### Official Review · AnonReviewer2 · 2019-11-03
**Official Blind Review #2**

**Rating:** 1

**Review:**

The paper is concerned with the presence of isomorphism bias in commonly used graph learning benchmarks. In particular, the paper analyzes the amount of isomorphic graphs in 54 graph datasets and evaluates the performance of three graph classification methods under two isomorphism settings.

Careful analyses of commonly used benchmarks can be important contributions that provide new insights into the performance of state-of-the-art models. The present paper's results on graph isomorphism properties can indeed be valuable for the ablation of models and testing their performance with regard to this property. I also found the relatively high label disagreements on some datasets (even under stronger isomorphism constraints) to be a surprising and useful result.

However, the main assumption of the paper -- which equates the quality of a graph learning benchmark with the amount of isomorphic graphs that it contains, i.e., the lower the better -- seems questionable.

The paper argues that isomorphic graphs are akin to duplicate images in computer vision and should be removed from a dataset. While completely identical graphs are certainly problematic, the case seems different for isomorphic graphs. In the latter, a learning method is required to identify the correct bijection form V_1 to V_2 which is a non-trivial task. Testing on isomorphic graphs evaluates the ability of a model to infer these equivalence classes from data which is an important property. Moreover, being able to capture the equivalence relation can be important for various graph learning tasks, e.g., to facilitate that two topologically equivalent graphs are be classified similarly.  Going back to the computer vision analogy: it seems a more adequate comparison for graph isomorphism would be translation and scale invariance which are certainly desirable properties for CV models.

In addition, the dataset analysis could also be improved. For instance, the SYNTHETIC dataset includes continuous node attributes that are essential for classification and make the graphs non-isomorphic (when considering, for instance, each attribute vector as a unique node label). However, the attributes are not considered in the analysis what leads to a large number of isomorphic graphs. On a side note: the paper also incorrectly attributes the SYNTHETIC dataset to (Morris et al, 2016), but it is in fact from [1]. The synthetic dataset of Morris et al (SYNTHIE) does not consist of isomorphic graphs, while the SYNTHETIC dataset of [1] does so intentionally.

The results of Section 5 seem also not very surprising: After removing node labels, it is expected that the number of isomorphic graphs increases since a discriminating feature has been removed. Moreover, when accounting for node labels, many standard benchmarks seem to consist of significantly less isomorphic and mismatched graphs (as can be seen in the appendix).

Since graph isomorphism != graph identity, the assumption Y_iso \sub Y_train in Property 6 seems also not appropriate. The results of Theorem 6.1 on the other hand seems straightforward and would hold for any classification task for which the true label for an equivalence class of instances is known.

**Experience Assessment:**

I have published in this field for several years.

**Review Assessment: Checking Correctness Of Derivations And Theory:**

I assessed the sensibility of the derivations and theory.

**Review Assessment: Checking Correctness Of Experiments:**

I assessed the sensibility of the experiments.

**Review Assessment: Thoroughness In Paper Reading:**

I read the paper at least twice and used my best judgement in assessing the paper.

---

> ### Author Response · Authors · 2019-11-06
> **Thank you for thorough review**
>
> First, we would like to make an important distinction between graph isomorphism and graph identity. In graph isomorphism, we are given purely topology of graphs and asked to find a bijection between nodes. In graph identity, we may also be given meta-information such as node labels, and need to find isomorphism that preserves labels. Graph identity is more general than graph isomorphism, but as we can see even using additional meta-information, we have a lot of identical graphs, which are problematic for correct validation. Thus our main contribution is the new clean data sets which do not have this problem.
>
> Regarding the main assumption. As we discussed, many models introduce isomorphism mechanisms into the models [1,2,3], which show success in the experiments. These mechanisms guarantee that the models have discriminative power to detect isomorphic graphs by comparing the embeddings of these graphs or a corresponding kernel value. If any such model can do it for any pair of graphs, then such a model would solve graph isomorphism, which is GI-hard, so no one has a guarantee on all graphs. But, many models have an approximation that works almost on all pairs of graphs. For example, WL kernel or GIN has expressive power the same as WL algorithm, which is known to work almost on all graphs [4], and definitely on all graphs that we encounter in graph classification data sets: they are just too easy for isomorphism testing. With that said, a property of distinguishing isomorphic graphs is an important one and is attained in polynomial time via WL algo or others (e.g. graphlet distribution, anonymous walks, etc.) on the data sets that everyone uses. Hence, all such models can easily exploit train-to-test leakage by just comparing embeddings or kernel values of a corresponding pair of graphs.
>
> Not only is there a leakage problem with current data sets, but also the performance is measured in an incorrect manner: the more isomorphic instances we have, the better performance of the model will be, but the expressivity of the model is not measured in this case. To see this, imagine two data sets: (1) with 10 different graphs and (2) with 20 isomorphic graphs. It is more important for a model to be good on the (1) data than on (2), because second measures the performance on essentially 1 graph. To remove these problems altogether, we propose a simple solution: remove graphs that can distort the validation metrics.
>
> Note that the problem is not fully resolved even if we use meta-data information, because (1) even with meta-information the number of isomorphic graphs is significant (80% of 40 data sets have isomorphic graphs even if we use node labels), (2) many data sets do not have meta-information, e.g. very popular data set IMDB-MULTI has 80.8% isomorphic graphs, while not having any additional meta-data.
>
> Regarding additional data set analysis. Indeed we are aware that SYNTHETIC data set includes continuous attributes. In fact, at the beginning of Section 5 we write:
> “... in Synthetic data set all graphs are topologically identical but the nodes are endowed with normally distributed scalar attributes ...”.  As discussed in Section 5, meta-information includes node/edge labels/attributes and taking all of this into account will increase the diversity of a data set.
> We will correct the reference for SYNTHETIC data set.
>
> Regarding Section 5. While these results may not be surprising, it is important to explain the found phenomenon: why there are so many isomorphic graphs. Indeed, the main reason is that meta-information matters to distinguish topologically equivalent graphs. Please note that it is not trivial to obtain label-preserving isomorphism of graphs and we had to write an efficient procedure to obtain isomorphic pairs with node labels. This is very time- and resource-consuming and even with an efficient code implementation it still takes hours to obtain all statistics for all data sets. One of our contributions is that we release the code so that anyone with a new data set can verify on isomorphism of graphs. We also note that two other reasons are the size size of graphs, which makes it unlikely to have an isomorphic data set, and origin of the data set (e.g. for IMDB networks it is very likely to have similar actors relationship, which dictates isomorphism of graphs).
> Regarding Property 6. Please note in the case of graph identity, we can also have $Y_{iso} \subset Y_{train}$, so the property still holds.
> Regarding Theorem 6.1. While it’s an easily derived fact, it is important to state this theorem as it explains a model-agnostic way to improve classification results.
>
> We hope this feedback is useful and you can reassess our paper. We would like to answer more questions if you have any.
>
> [1] “Weisfeiler-Lehman Graph Kernels” 2012
> [2] “Anonymous Walk Embeddings” 2018
> [3] “How Powerful are Graph Neural Networks?” 2019
> [4] “On the Combinatorial Power of the Weisfeiler-Lehman Algorithm” 2017

---

### Official Review · AnonReviewer4 · 2019-11-04
**Official Blind Review #4**

**Rating:** 3

**Review:**

This work probes graph classification datasets for isomorphism bias. They find substantial amount of bias in some datasets and show that they suffer from data leakage. They further perform a more fine-grained evaluation taking into consideration the node/edge types which reduce the perceived effects. They also provide some recommendations for measuring the 'right metrics' and release clean versions of the considered datasets.

Strengths:
- The methodology is rigorous and the datasets considered is extensive
- The paper is well written

Concerns:
- Isomorphism is not necessarily a bad thing in graph classification tasks. Especially in chemistry where a bond decides if a compound is poisonous or not. Also, as the authors themselves mention, taking node/edge labels decrease the isomorphism in most datasets.
- The results and recommendations presented in the paper are intuitive and somewhat trivial
- I am not sure if ICLR is the right venue for this work

**Experience Assessment:**

I have published in this field for several years.

**Review Assessment: Checking Correctness Of Derivations And Theory:**

I carefully checked the derivations and theory.

**Review Assessment: Checking Correctness Of Experiments:**

I carefully checked the experiments.

**Review Assessment: Thoroughness In Paper Reading:**

I read the paper thoroughly.

---

> ### Author Response · Authors · 2019-11-05
> **Thank you for the review**
>
> Since the area of graph classification is rapidly growing (with several submissions just in ICLR 2020 that use these data sets), it’s important to set the right rules for correct validation of newly proposed models. As we described in the related work section, such a problem is already the case for Computer Vision data sets, which can severely skew the validation accuracy and we want to prevent this for graph classification domain.
>
> The fact that so many data sets contain so many isomorphic graphs is a newly discovered phenomenon and is novel to the community. While there were some mentions [1] about this fact, no one has run a full analysis of the data sets, for obvious reasons: it’s very time- and resource-consuming problem. It’s not trivial to find all isomorphic pairs in 54 data sets: for example for the largest data set, there are more than 71 million graph pairs that need to be checked on isomorphism. Isomorphism test is known to be a challenging task and we had to write efficient code to preprocess pairs for this task and yet it still takes hours to compute statistics for all data sets. One of our contributions is that we release code and anyone with a new data set can easily run a pairwise isomorphism test on it.
>
> Regarding your first concern. Isomorphism in itself is not a bad thing and incorporating isomorphism features into the models has proven to lead to better results. The main issue of having isomorphic graphs in the data sets is that the model can “memorize”, explicitly or implicitly, the target labels during the training phase; this is obviously a leakage for validation and artificially increases the accuracy as we show in Table 4. It’s important to note here that if a model can memorize, it can also solve graph isomorphism, which is GI-hard and no one has claimed such model yet. But, many successful models [2,3,4] incorporate the features efficiently (e.g. via WL algo), which solves isomorphism problem for almost all pairs of graphs, including all pairs in used graph data sets. That means that such models can essentially memorize which graphs were seen in the training, by comparing test graph embedding with train graph embeddings. Now, imagine increasing the data set size by adding isomorphic graphs. Then, no matter how _weak_ classification model is, the accuracy on the increased data set will be approaching 100%, by just memorizing the labels from the train set and all models would be the same in such data sets. This is not the goal for the graph classification problem, which rather asks to find a model that would be correct on as large set of new instances as possible. Informally, comparing performance on a data set of 100 different graphs is better than on a data set of 1000 isomorphic graphs. That's why having clean data sets without isomorphic instances has more fair validation comparison.
>
> Regarding using node/edge labels for the models, indeed it reduces the number of isomorphic graphs, however (1) many data sets still contain isomorphic graphs as we show in Figures 3,4,5; (2) some models do not use node/edge attributes; and (3) not all data sets have such meta-data (e.g. IMDB-MULTI contains 80% of isomorphic graphs, but does not have any attributes). Moreover, this is a new insight from our work as to why it is important to use _given_ node/edge attributes when designing a model, instead of designing attributes heuristically, as it has been tried before.
>
> Regarding your second concern. Having a paper intuitive is not necessarily a bad thing. We kept in mind that it should be an easy-to-read work so that more graph practitioners could use the right framework (data sets, meta-data, validation procedures, etc.) and these hidden parts should be known to all who want to design a model. While property and theorem are easily derived facts, it is important to state them as they partially explain the current performance of classification models.
>
> Regarding your last concern, please note that our analyzes encompasses almost all datasets currently used for graph classification benchmarking. This area has been dominant for representation learning of graphs, leading to advancements in Graph Neural Networks research, and thus a clean data set is a prerequisite for learning a good graph embedding. We also note that many of the data sets come from the Biological application, one of the ICLR main applications (as denoted in the main website) and some results of this paper provide new insights into this application (e.g. noisy target labels for many graphs; necessity to use node/edge attributes; etc.).
>
> We hope this feedback addresses your concerns and we hope it can lead to your reevaluation of this paper. If you have further questions, please let us know.
>
> [1] “A Survey on Graph Kernels” 2019
> [2] “Weisfeiler-Lehman Graph Kernels” 2012
> [3] “Anonymous Walk Embeddings” 2018
> [4] “How Powerful are Graph Neural Networks?” 2019

---

### Author Response · Authors · 2019-11-15
**Changes log**

Dear Reviewers and Area Chairs,

We carefully reviewed the concerns and we strengthened our paper with the theoretical part (for convenience, in red color in an updated version of the paper).

We theoretically derive an upper bound for the generalization gap expressed in the terms of Rademacher complexity of a classifier and the number of isomorphic graphs in a data set. Intuitively, Rademacher complexity measures the expressiveness of the model. Then, the classification problem with isomorphic graphs can be interpreted as a classification with a weighted loss. We prove the following upper bound:

$$\sup_{f\in\mathcal{F}}\Bigl(\mathbb{E}_{\mathbb{P}}l(f(x),y) -\mathbb{E}_{\mathcal{D}}l(f(x),y)\Bigr) \leq G(w) + \mathcal{R}_N(\mathcal{F}),$$

where $\mathcal{R}_N(\mathcal{F})$ is the Rademacher complexity of a function class $\mathcal{F}$ and $G(w)$ is the function of the weight parameter that corresponds to the class imbalance.

By optimizing the RHS of this upper bound and finding optimal value $w_{opt}$ we show that the conditions for minimal upper bound, hence making a generalization gap tighter. We note that in the imbalanced case, for the optimal value of $w_{opt}$, expressed via class imbalance $\pi$, the generalization gap tends to zero as the number of graphs $N$ grows. Hence any deviation from the optimal value of the number of isomorphic graphs $w_{opt}$, which evidently happens in real-world data sets, may result in overfitting of the model. This theoretical result is supported in the experiments in Table 4.

With this result, not only do we demonstrate the problems with the current data sets used ubiquitously for graph representation, but also show a theoretical support why the generalization abilities of the classifiers can be limited.

---

### Public Comment · ~Chen_Cai1 · 2020-02-08
**A relevant paper on graph classification**

Hello,

It's nice to see a detailed analysis of the graph classification dataset. I would like to point out a relevant paper on using simple node-degree features for graph classification. On many popular non-attributed datasets, our method yields comparable results with the many popular graph networks and graph kernels. Better datasets are definitely needed for evaluating different methods.

https://arxiv.org/abs/1811.03508
Cai, Chen, and Yusu Wang. "A simple yet effective baseline for non-attributed graph classification." arXiv preprint arXiv:1811.03508 (2018).

---

### Decision · Program_Chairs · 2019-12-19

**Decision:**

Reject

**Comment:**

Thanks to reviewers and authors for an interesting discussion. It seems the central question is whether learning to identify correct bijections should be part of graph classification problems, or whether this leads to bias and overfitting. Reviews are generally negative, putting this in the lower third of the submissions. The paper, however, inspired an interesting discussion, and I would encourage the authors to continue this line of work, addressing the question of bias and overfitting more directly, possibly going beyond dataset evaluation and, for example, thinking about how to evaluate whether training on non-isomorphic graphs leads to better off-training set generalization.